
# Aerosol properties and meteorological conditions in the city of Buenos Aires, Argentina during the resuspension of volcanic ash from the Puyehue-Cordón Caulle eruption

A. G. Ulke[1,2], M. M. Torres Brizuela[1], G. B. Raga[3], and D. Baumgardner[4]

[1]Departamento de Ciencias de la Atmósfera y los Océanos, Facultad de Ciencias Exactas y Naturales, Universidad de Buenos Aires, Argentina
[2]Unidad Mixta Internacional (UMI) – Instituto Franco Argentino sobre Estudios de Clima y sus Impactos (IFAECI)/CNRS, Buenos Aires, Argentina
[3]Centro de Ciencias de la Atmósfera, Universidad Nacional Autónoma de México, DF, México
[4]Droplet Measurement Technologies, Boulder, CO, USA

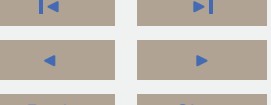
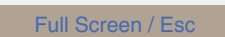


Received: 10 November 2015 – Accepted: 11 December 2015 – Published: 15 January 2016

Correspondence to: A. G. Ulke (anagulke@gmail.com)

Published by Copernicus Publications on behalf of the European Geosciences Union.

Discussion Paper | Discussion Paper | Discussion Paper | Discussion Paper |

**NHESSD**

doi:10.5194/nhess-2015-311

**Aerosol properties and meteorological conditions in the city of Buenos Aires**

A. G. Ulke et al.

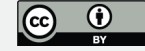

## Abstract

The eruption in June 2011 of the Puyehue-Cordón Caulle Volcanic Complex in Chile impacted air traffic around the Southern Hemisphere for several months after the initial ash emissions. The ash deposited in vast areas of the Patagonian steppe was subjected to the strong wind conditions prevalent during the austral winter and spring, experiencing resuspension over various regions of Argentina.

In this study we analyze the meteorological conditions that led to the episode of volcanic ash resuspension, which impacted the city of Buenos Aires and resulted in the closure of both airports on 16 October 2011. The thermodynamic soundings show the signature of "pulses of drying" associated with the presence of hygroscopic ash in the atmosphere that has been reported in similar episodes after volcanic eruptions in other parts of the world.

Measurements of aerosol properties that were being carried out in the city during the resuspension episode indicate the presence of an enhanced concentration of aerosol particles in the boundary layer. Reports of ash on the runway at the airport near the measurement site correlate in time with the enhanced concentrations. Since the dynamics of ash resuspension and recirculation is similar to the dynamics of dust storms, we use the HYSPLIT model with the dust storm module to simulate the episode that affected Buenos Aires. The results of the modeling agree qualitatively with satellite lidar measurements.

## 1   Introduction

After an inactive period of ca. 50 years the Puyehue-Cordón Caulle Volcanic Complex (PCCVC) in Chile (40.59° S, 72.11° W, 2.236 m a.s.l.) erupted on 4 June 2011, ejecting a plume higher than 12 km a.s.l. The earthquake activity and eruption events were monitored by the Chilean agency on geology and mining policies, Servicio Nacional de Geología y Minería (SERNAGEOMIN), which provides the periodic status reports

Discussion Paper | Discussion Paper | Discussion Paper | Discussion Paper |

# NHESSD

doi:10.5194/nhess-2015-311

**Aerosol properties and meteorological conditions in the city of Buenos Aires**

A. G. Ulke et al.

**NHESSD**

doi:10.5194/nhess-2015-311

**Aerosol properties and meteorological conditions in the city of Buenos Aires**

A. G. Ulke et al.

of Chilean volcanoes. A detailed chronological description of the volcanic eruption and modeling of the volcanic ash dispersion during the main phase explosive events are reported by Collini et al. (2013). As a result of this eruption many airports were closed and flights cancelled in Chile, Uruguay and Argentina because of the potentially hazardous operating conditions. The PCCVC eruption also impacted air traffic throughout the Southern Hemisphere for several days from South America to Oceania, as prevailing winds at higher levels (300–600 hPa) over this region are strong and westerly.

The eruption of the PCCVC occurred during an extended field campaign when measurements of the properties of atmospheric aerosol particles were being made in Buenos Aires to characterize pollution in the city (Ulke et al., 2011; Raga et al., 2013). Four case studies were analyzed and evaluated that were coincident with the arrival of the ash plume over the Buenos Aires area and that led to airport closures. Vertical profiles of aerosol backscatter, measured with a ceilometer, clearly identified the presence of the volcanic ash.

It is well documented that volcanic ash can be hazardous to flight operations because ash particles have silicate compounds that can reach their melting point in the jet engine turbines and subsequently crystallize on the turbine blades. This leads to costly and sometimes deadly damage (Casadevall, 1993, 1994). However, fresh volcanic ash emissions are not the only situations that can be hazardous. The ash deposited at the surface from previous eruptions can be resuspended by winds and advected to other regions, as has been reported in various studies. In particular, Hadley and Hufford (2004) reported an episode of resuspended ash in Alaska during September 2003, and observed that the plume of lofted ash reached an altitude of over 1600 m and extended 230 km into the Gulf of Alaska. Leadbetter et al. (2012) modeled resuspended ash in case studies after the Eyjafjallajökull eruption using NAME (Numerical Atmospheric-dispersion Modeling Environment), the UK Met Office's Lagrangian particle dispersion model (Jones et al., 2007) and found good agreement with $PM_{10}$ observations and satellite RGB dust product images. The dynamics of the resuspension of volcanic

ash is quite similar to dust storm episodes and the resuspension of volcanic ash has been widely studied and modeled as dust storms (e.g. Gillette and Passi, 1988; Freudenthaller, 2006; Gu et al., 2003; Jugder et al., 2011; Dupart et al., 2012).

An episode of resuspension associated with the PCCVC activity occurred in Argentina from 14 to 17 October 2011, which impacted air traffic around Buenos Aires on 16 October where all flight operations were cancelled by the national authorities (Secretary of Transport). Also the international airport in Montevideo, Uruguay was closed for flight operations on 17 October with reports of ash deposited on the runways. Measurements were still being made in the aforementioned field campaign during this period and provide a unique opportunity to assess the impact of the resuspended ash and to evaluate models used to predict resuspension events. Folch et al. (2014) performed numerical simulations of the episode reported in this paper with the FALL 3-D model, focusing on the calibration of a variety of model parameters and treatments, including the resuspension process of ash deposited at the surface.

The objectives of this paper are:

– To analyze the near surface and tropospheric thermodynamic and kinematic conditions that led to the resuspension of deposited volcanic ash into the atmospheric boundary layer and its transport towards the Buenos Aires area.

– To assess changes in the atmospheric moisture content and temperature of the thermodynamic vertical profiles due to the presence of the ash plume.

– To evaluate changes in the aerosol properties in Buenos Aires during the resuspension event.

– To evaluate the performance of the Hybrid Single-Particle Lagrangian Integrated Trajectory (HYSPLIT) (Draxler and Hess, 1998) model as a tool to predict ash resuspension.

Discussion Paper | Discussion Paper | Discussion Paper | Discussion Paper |

**NHESSD**

doi:10.5194/nhess-2015-311

**Aerosol properties and meteorological conditions in the city of Buenos Aires**

A. G. Ulke et al.

## 2 Measurements and methodology

### 2.1 Geographical location

The ash from the eruptions of the PCCVC (40.59° S, 72.11° W, 2.236 m a.s.l., indicated by the number 1 in Fig. 1a) in 2011 were deposited predominantly downwind in Argentina, in an arid region known as the Patagonian steppe. The bare soil in this arid region, combined with the prevailing and strong westerly winds, promotes the occurrence of dust storms. It is worth mentioning that the climatological analysis (SMN Climatological reports, accessed at http://www.smn.gov.ar/serviciosclimaticos/ on December 2011) indicated a dry period during 2011. The city of Buenos Aires, which was affected by the ash from the PCCVC, is indicated by the number 2 in Fig. 1a. The resuspended volcanic ash is seen in the Moderate Resolution Imaging Spectroradiometer (MODIS), 1 km resolution visible data on 15 October in Patagonia (Fig. 1b). A technical report from the Instituto Nacional de Tecnología Agropecuaria (Gaitan et al., 2011) based on field measurements depicts the deposition zone (isopachs) in the same area.

### 2.2 Instrumentation at the field site and datasets

A detailed description of the instrumentation deployed at the Ciudad Universitaria (34°35′ S, 58°22′ W) during the field campaign to evaluate urban aerosol pollution is presented by Ulke et al. (2011) and Raga et al. (2013) and is only briefly described here. The particle properties measured were the number concentration of condensation nuclei (CN) larger than approximately 50 nm, the mass concentration of particle-bound polycyclic aromatic hydrocarbons (PPAH), the scattering ($B_{scat}$) and absorption ($B_{abs}$) coefficients at 530 nm and the vertical profile of backscattered light from aerosols at a wavelength of 910 nm. For the purpose of studying urban pollution, the measurements of $B_{abs}$ are converted to an equivalent mass concentration of black carbon (eBC) using a specific mass absorption coefficient of $6.5\,\mathrm{m^2\,g^{-1}}$ (Bond and

Discussion Paper | Discussion Paper | Discussion Paper | Discussion Paper | Discussion Paper |

**NHESSD**

doi:10.5194/nhess-2015-311

**Aerosol properties and meteorological conditions in the city of Buenos Aires**

A. G. Ulke et al.

Bergstrom, 2006). The eBC concentration is an indicator of emissions from combustion by diesel engines, as are common in buses and trucks in Buenos Aires. Dust particles and volcanic ash (not only black carbon) can absorb radiation at 530 nm. However, the presence of dust and volcanic ash will typically not produce a PPAH signal, so the measurements can provide evidence of light absorbing particles that are not originated by diesel engines. The ceilometer backscatter measurements were converted to extinction coefficients using a least squares analysis with the in situ measurements of $B_{scat}$ and $B_{abs}$ (Raga et al., 2013). The vertical resolution of the ceilometer is 20 m and the measurements are 10 min averages.

Additional information on aerosol optical properties was obtained from the AERosol RObotic NETwork (AERONET) sun photometer (Holben et al., 1998), maintained and operated by personnel at Centro de Investigaciones en Láseres y Aplicaciones (CEILAP-BA 34°39′35″ S; 58°28′08″ W), located approximately four kilometers west of the Ciudad Universitaria measurement site. Although the AERONET sun-photometer measures at multiple wavelengths, only the aerosol optical depth (AOD) at 500 nm was selected for analysis. This wavelength is closest to the 530 nm wavelength used in the instruments that measured the $B_{scat}$ and $B_{abs}$ and the 550 nm used by MODIS that is carried on the Aqua (Parkinson, 2003) and Terra satellites. MODIS images were used to identify the resuspension events.

Measurements obtained by the Cloud-Aerosol Lidar and Infrared Pathfinder Satellite Observations (CALIPSO) are also analyzed. The Cloud-Aerosol Lidar with Orthogonal Polarization (CALIOP) is the primary instrument on the CALIPSO satellite. CALIOP acquires vertical profiles of elastic backscatter at two wavelengths (1064 and 532 nm) from a near nadir-viewing geometry during both day and night phases of the orbit. In addition to the total backscatter at the two wavelengths, CALIOP also provides profiles of linear depolarization at 532 nm. The detection limit is determined by a threshold setting in the layer detection algorithm (Vaughan et al., 2009). At night this threshold is about $2 \times 10^{-4}$ km$^{-1}$ sr$^{-1}$ in the mid- to lower-troposphere and during daytime, is a factor of 2 to 5 less sensitive.

**NHESSD**

doi:10.5194/nhess-2015-311

**Aerosol properties and meteorological conditions in the city of Buenos Aires**

A. G. Ulke et al.

Discussion Paper | Discussion Paper | Discussion Paper | Discussion Paper |

The profiles are used by a scene classification algorithm composed of three submodules: cloud–aerosol discrimination (CAD) (Liu et al., 2009), aerosol subtyping (Omar et al., 2009), and cloud ice-water phase discrimination (Hu et al., 2009). Only two classes of tropospheric features are defined in the CAD algorithm: "cloud" and "aerosol". The first class includes clouds, fogs and mists, whereas hazes belong to the aerosol class. The aerosol classification algorithm only operates on those inputs from the CAD algorithm that have been classified as aerosols. Six aerosol types are defined: clean continental, clean marine, dust, polluted continental, polluted dust, and smoke. To determine aerosol type, these algorithms use the integrated attenuated backscatter and the volume depolarization ratio measurements, as well as surface type and layer altitude. The volume depolarization ratio is used to identify aerosol types with a substantial mass fraction of aspherical particles (e.g., a mixture of smoke and dust). The integrated attenuated backscatter is used to discern instances of transient high aerosol loading over surfaces where this is not usually expected. The total attenuated color ratio is an independent quantity because it is not used in the subtyping algorithm.

Although the standard processing algorithms do not attempt to identify material as volcanic, CALIOP observations have been used to study the plumes from a number of volcanic eruptions. Polarized lidar backscatter signals can be used to discriminate between ash and sulfate aerosol resulting from volcanic emissions because light scattered from spherical particles, such as sulfate aerosol droplets, retain the linear polarization of the incident light, whereas backscatter from irregular solid particles, like ash, is depolarized (Winker et al., 2012; Vernier et al., 2013).

According to Winker et al. (2012), about half the ash layers from Eyjafjallajökull were classified as cloud, and otherwise as either "desert dust" or "polluted dust", due to the depolarization signature of the ash. They identified the layers of volcanic origin by manual inspection of backscatter, volume depolarization, and attenuated color ratio browse images and also considered the altitude and layer morphology to help distinguish the plumes from cirrus, desert dust and boundary layer aerosols. Based on similarities found by Schumann et al. (2011) between the measured properties (size

**NHESSD**

doi:10.5194/nhess-2015-311

**Aerosol properties and meteorological conditions in the city of Buenos Aires**

A. G. Ulke et al.

distribution, composition, shape) of transported Eyjafjallajökull ash and desert dust, Winker et al. (2012) assumed that ash plumes observed after transport of more than one day will have optical properties similar to that of transported desert dust. Aiming to improve volcanic ash warnings for aviation safety, Vernier et al. (2013), developed an ash detection algorithm to discern low volcanic ash loadings and to describe the three-dimensional structure of the ash cloud. The method is based on optical properties (depolarization and color ratios) of the ash plume from the PCCVC located near 6–13 km over Australia and New Zealand and observed by the CALIPSO lidar. Ash cloud observations ingested into a Lagrangian model were used to construct ash dispersion maps and cross sections.

The government of the city of Buenos Aires operates three air quality monitoring stations that report hourly concentrations of the coarse fraction of aerosol particles with aerodynamic diameters smaller than 10 μm ($PM_{10}$), according to US-EPA regulations. The data from the station La Boca were selected since the station is close to the river and has similar geographic characteristics as the Ciudad Universitaria site. The data were digitized from the graphs displayed on the government web page: http://www.buenosaires.gob.ar/areas/med_ambiente/apra/calidad_amb/red_monitoreo/index.php?menu_id=34232.

The surface data from synoptic stations located at the Buenos Aires airports, Ezeiza and Aeroparque, and also METAR, SPECI, TAF, SIGMET reports and upper-air observations at 12:00 UTC (09:00 LST) were provided by the SMN. An extended analysis was performed on the radiosondes launched from the airports in Ezeiza (34°49′ S; 58°32′ W) and Santa Rosa (36°34′ S; 64°16′ W). Santa Rosa represents the northern limit of the Patagonian steppe and Ezeiza is the closest to the experimental site (see Fig. 1b).

The Global Data Assimilation System (GDAS) at 1° × 1° resolution of the National Center for Environmental Prediction (NCEP) meteorological data were used to analyze the meteorological conditions and to drive the numerical model.

**NHESSD**

doi:10.5194/nhess-2015-311

**Aerosol properties and meteorological conditions in the city of Buenos Aires**

A. G. Ulke et al.

Discussion Paper | Discussion Paper | Discussion Paper | Discussion Paper

## 2.3  Modeling approach

The HYSPLIT model (available at http://www.arl.noaa.gov/ready/hysplit4.html), that computes trajectories and dispersion using either a puff or particle approach was selected to model the ash resuspension event analyzed here. Note however, that the

5 HYSPLIT model is only used as a qualitative tool as there were no data available on the amount, characteristics or areal extent of deposited ash to compare against the modeling results. Backward trajectories from the city of Buenos Aires were computed to confirm that the source of the air mass with a high content of volcanic ash that reached the city on 16 October was indeed the region shown in the triangle in Fig. 1. Several

sensitivity tests were performed with the model to determine the optimum setup.

As mentioned in the introduction, events of resuspension of volcanic ash are dynamically similar to dust storms, so the lifting and subsequent advection of the ash particles can be simulated using the dust emission algorithm included in the current public version of HYSPLIT, which has been previously evaluated for several wind-blown

dust emissions (Draxler et al., 2010).

The "dust storm" option of HYSPLIT uses the concept of a threshold friction velocity dependent on surface roughness and soil type. A pre-processor for "desert" soil type identifies a dust emission cell and an emission rate is then computed when the local wind speed exceeds the threshold set for the particular soil characteristics

(Draxler et al., 2013). Analysis of the preliminary results revealed an erroneous representation of the land type, since the file with the original soil type not only has a rather coarse resolution (1°) but also misrepresents the soil type in the Patagonian region. This error in classification stems from the fact that the file containing soil characteristic is not frequently updated (e.g. every 15 days in the

model NAME). Soil type files should be dynamically coupled, incorporating soil moisture and soil-type changes such as those that occur in the case of ash deposits, as mentioned in Leadbetter et al. (2012). In order to overcome this problem, the default land use file was modified to better represent the soil type in the Patagonian

Discussion Paper | Discussion Paper | Discussion Paper | Discussion Paper |

**NHESSD**

doi:10.5194/nhess-2015-311

**Aerosol properties and meteorological conditions in the city of Buenos Aires**

A. G. Ulke et al.

steppe (desert soil). Hence, an important contribution of the present study is the update of the land use database for Argentina, based on the dataset provided by the Food and Agricultural Organization (http://www.fao.org/soils-portal/soil-survey/soil-maps-and-databases/faounesco-soil-map-of-the-world/en/).

⁵ Several numerical experiments simulating the dust storm were performed, utilizing the Khantar–Clayson turbulence parameterization in the vertical and horizontal turbulence proportional to the vertical. A total of 45 000 particles were resuspended at each grid cell where the desert soil type is found and when the threshold wind velocity is exceeded. Since few measurements exist of the particle characteristics in

¹⁰ Patagonia, the number of particles was chosen according to the concentrations at Gobi desert regions of Mongolia and North China reported by Dupart et al. (2012). Runs were initialized on 13 October 2011 at 18:00 UTC and lasted for 96 h.

## 3  Analysis of the meteorological conditions

### 3.1  Vertical profiles

¹⁵ The thermodynamic evolution during this case study is illustrated by the vertical profiles shown in Fig. 2. The Santa Rosa (number 3 in Fig. 1) and Ezeiza soundings (number 2 in Fig. 1) are analyzed for 14 through 16 and 15 to 17 October, respectively, following the northeastward displacement of the volcanic plume. These are compared with the mean decadal soundings (red curves) obtained by Velasco and Necco (1980).

²⁰ The Santa Rosa thermodynamic profiles exhibit a warm layer located from the surface to ∼ 1500 m compared to the mean temperature profile (Fig. 2a), throughout the period analyzed. Note that initially the near-surface temperature inversion is shallow (compared to the mean) and weakens with time. This evolution is mainly driven by the stronger radiative heating of the surface (almost bare soil covered with volcanic ash),

²⁵ not by advection or turbulent mechanical mixing. Above this layer, the quasi-isothermal profile evolves into an adiabatic lapse rate. The atmospheric humidity profile depicts

Discussion Paper | Discussion Paper | Discussion Paper | Discussion Paper | Discussion Paper |

**NHESSD**

doi:10.5194/nhess-2015-311

**Aerosol properties and meteorological conditions in the city of Buenos Aires**

A. G. Ulke et al.

**NHESSD**

doi:10.5194/nhess-2015-311

**Aerosol properties and meteorological conditions in the city of Buenos Aires**

A. G. Ulke et al.

layers of drying likely associated with the presence of a mixture of silica and dust in the air (Lathem et al., 2011). Schumann et al. (2011) and Miffre et al. (2012) also reported this atmospheric behavior in their analysis of the Eyjafjallajökull eruption. The relevant dynamic features of the vertical profiles are: (i) between 14 and 15 October the wind speed is accelerating above 300 m hence increasing in the potential to resuspend and transport the ash and (ii) veering of the wind from west to southwest throughout the free troposphere during 15 and 16 October (see Fig. 2a).

The Ezeiza thermodynamic profiles also exhibit warmer conditions than the climatological mean values (Fig. 2b) up to $\sim 1.5$ km height during the first 48 h; however, on 17 October 2011, the temperature is lower coinciding with the northeastward movement of the high pressure system that covers northern Patagonia and Central Argentina (as can be seen in Fig. 4). The humidity profiles show a dryer atmosphere than the mean decadal values (Fig. 2b) and this feature is more noticeable on 16 October, which shows a dry layer from the surface up to 1.8 km. The layer from the surface up to 400 m experienced an extreme drying (from $8.5\,\mathrm{g\,kg^{-1}}$ down to $3.5\,\mathrm{g\,kg^{-1}}$) also associated with static stability, and coinciding with the presence of resuspended material over Buenos Aires. This dry layer in the thermodynamic sounding manifests itself as a layer with the highest extinction values ($\sim 100\,\mathrm{Mm^{-1}}$) in the ceilometer measurements obtained at the research site, as seen in Fig. 3. The wind speed at the surface decreased from $5\,\mathrm{m\,s^{-1}}$ to mostly calm during this period (Fig. 2b), contributing to stagnation and increased ambient concentration of the resuspended volcanic ash (Fig. 3). The highly variable extinction coefficient throughout 16 October reflects the inhomogeneity of the ash cloud and the effect of thermal turbulence driven by the surface radiative surplus. By 17 October the larger scale circulations have moved the ash cloud away from the Buenos Aires region as is evident in the temporal evolution of the extinction coefficient (see Fig. 3).

Discussion Paper | Discussion Paper | Discussion Paper | Discussion Paper |

## 3.2 Synoptic horizontal analysis and surface observations

The analysis of the relevant synoptic-scale features every 6 h was performed for several days prior to the resuspension event, but only the times that coincide with the radiosonde launches shown in Fig. 2 are presented here.

On 15 October at 12:00 UTC an anticyclonic system was centered at about 32° S, 95° W accompanied by a low-pressure system in southern Patagonia (Fig. 4). The associated horizontal pressure gradient produces strong near surface southwestern winds as far as ∼ 40° S, capable of lifting the deposited ash. The contours of the 1000/500 thickness reveal a relatively warm air mass in northern Patagonia (Fig. 4a). North of 40° S near-surface winds were weak, in accordance with the upper air measurements. The equivalent potential temperature ($\theta_e$) at 850 hPa is used to follow the suspended ash and the value ∼ 305 K is considered indicative of the air mass which is restricted to an undulating region located between 42 and 35° S, enclosed with the black box in Fig. 4b. The location of this air mass coincides with an area of anticyclonic vorticity at 500 hPa (not shown) indicating a stable synoptic environment.

On 16 October at 12:00 UTC the contour of $\theta_e \approx 305$ K is observed over the northern area of Buenos Aires, as marked with the black box in Fig. 4d, coincident with the warm layer represented by the 500/1000 thickness (Fig. 4c). Buenos Aires is located between a low-pressure system over the Atlantic Ocean and an anticyclonic system to the west (Fig. 4d). The associated airflow shows a strengthening of near-surface winds that advect the ash towards the region of Buenos Aires. The presence of volcanic ash was reported on 16 October by the METAR/SPECI. The Aeroparque airport reported the presence of ash at 08:00 UTC and Ezeiza and Montevideo (Uruguay) airports documented it at 13:00 UTC. All airports in the area were closed for flight operations during the whole day due to the presence of volcanic ash not only in the atmosphere but that had also deposited on the runways.

During the first twelve hours of 17 October, the high-pressure system extended to the Atlantic Ocean (≈ 40° W) and, in the area of the city of Buenos Aires, the near-surface

Discussion Paper | Discussion Paper | Discussion Paper | Discussion Paper | Discussion Paper |

**NHESSD**

doi:10.5194/nhess-2015-311

**Aerosol properties and meteorological conditions in the city of Buenos Aires**

A. G. Ulke et al.

winds became easterly as shown in Fig. 4e. The $\theta_e$ values of 305 K are displaced to the north of Buenos Aires (Fig. 4f), and the city of Buenos Aires was no longer under the influence of the volcanic ash.

## 4  Analysis of the measurements from the field campaign

The objective of the continuous measurements obtained with the instrumentation deployed at the research site in Ciudad Universitaria was to evaluate the properties of ambient aerosols associated with urban emissions. Nevertheless, the measurements can also help to identify the presence of non-urban aerosol during the episodes of volcanic ash. In particular, the different correlations between the parameters measured provide insight into the origin of the aerosols sampled. Figure 5 shows time series of the PPAH and eBC (Fig. 5a) and the CN observed at the research site and $PM_{10}$ recorded at the La Boca station (hourly values) shown in Fig. 5b. Note that during the mornings of 15 and 17 October the PPAH, eBC and CN exhibit the typical behavior associated with emissions from urban traffic, reaching a maximum before noon. Concentrations of eBC and PPAH are well correlated, indicative of a common source (diesel combustion). However, in the late evening of 15 October the eBC concentration is elevated while the PPAH remains low suggesting a non-urban source (indicated by the black oval in Fig. 5a). The fact that eBC and PPAH are not correlated during this nighttime period suggests that the elevated absorption coefficient (from which eBC is estimated) is related to particles that are not typical of urban pollution. Ash and dust particles also extinguish radiation at 530 nm, so the signal of elevated eBC (determined optically) is most likely ash and dust, not black carbon, as explained in Raga et al. (2013). The CN values (Fig. 5b) are somewhat higher than the usual urban concentrations during the night, deviating from the typical diurnal cycle associated with urban traffic, as can be seen in Fig. 4 in Raga et al. (2013). The meteorological conditions and the ceilometer measurements (see Fig. 3) indicate the arrival of the resuspended ash around midnight on 15 October and the early morning hours of 16 October (DOY 289). These elevated

Discussion Paper | Discussion Paper | Discussion Paper | Discussion Paper | Discussion Paper |

**NHESSD**

doi:10.5194/nhess-2015-311

**Aerosol properties and meteorological conditions in the city of Buenos Aires**

A. G. Ulke et al.



values are then supplemented by the typical morning traffic emissions in the city. The particles during this morning are very likely a mix of urban pollutants, aged ash and maybe some dust. All the monitoring network sites in the city show increased concentrations of $PM_{10}$, and the La Boca station shows values in excess of $240\,\mu g\,m^{-3}$ (for the daily average), much larger than the EPA standard of $150\,\mu g\,m^{-3}$. Note that the $PM_{10}$ concentration at La Boca, located $\sim 13\,km$ south of the research site, starts to increase earlier on the evening of 15 October, consistent with the resuspended ash plume arriving into the city. The average value of $PM_{10}$ at this station for all other days during the month of October is only about $40\,\mu g\,m^{-3}$. The hourly maximum of $PM_{10}$ is observed between midnight 15 October and 1 a.m. 16 October (see Fig. 5b), a time not associated with high pollution, and is further evidence of the presence of the aged ash in Buenos Aires. These in situ measurements can be related to the extinction coefficient derived from the ceilometer between 15 and 17 October 2011 shown in Fig. 3. Note that on 15 October the small values of the extinction coefficient are indicative of a clean air mass over the measurement site. In contrast, large backscatter signals denote the presence of the volcanic ash the early hours of 16 October and persisting for many hours. During nighttime and early morning hours, aerosols were present in a layer that extended $\sim$ up to $500\,m$. Recall from the results in the previous section that the wind was calm in Buenos Aires during this day, leading to higher ambient concentrations of particles with a large extinction signal. The development of the convective boundary layer is evident from the ceilometer data after local noon, with aerosols reaching a mean height of $1000\,m$ late in the afternoon, and the top of the plume of aerosols at $2000\,m$. The ceilometer data only captures the vertical profile of aerosols above the research site, and on the day of the arrival of the resuspended ash, it appears that the plume is highly non-uniform with different "filament-like" plumes contributing to enhance the near surface concentrations throughout most of the day. In the evening of 16 October, near the surface the in situ measurements indicate much lower concentrations of aerosol particles, in agreement with the $PM_{10}$ values.

Discussion Paper | Discussion Paper | Discussion Paper | Discussion Paper | Discussion Paper |

**NHESSD**

doi:10.5194/nhess-2015-311

**Aerosol properties and meteorological conditions in the city of Buenos Aires**

A. G. Ulke et al.

Surface winds at Aeroparque on 15 October are from the NW sector ranging from 2 to 6 m s$^{-1}$, with the maximum strength at 15:00 LST. Calm conditions in the early morning hours of 16 October are followed by light winds (2 m s$^{-1}$) with variable directions. An anticlockwise veering is observed from SW to N till 15:00 LST. Afterwards an important wind strengthening occurred (reaching $\sim 7.5$ m s$^{-1}$) along with a clockwise turning consistent with the entrance of the high pressure system previously mentioned. The increase of the surface wind promotes turbulent mixing and the resulting enhancement of the mixing layer height (see Fig. 3). The next day wind comes from the SE sector with the highest intensities ($\sim 7$ m s$^{-1}$) at the beginning of the day and the least ($\sim 2$ m s$^{-1}$) around midday in agreement with the highest extinction magnitudes registered by the ceilometer (see Fig. 3).

While the in situ sensors at the research site do not provide direct evidence of aerosol composition, the elevated concentrations observed on 16 October can indeed be identified as ash, since independent METAR/SPECI reports indicate the presence of ash at the Aeroparque airport located only 1 km away from the research site. Moreover, Level 2 AERONET measurements from CEILAP-BA indicate a significant contribution of the fine mode to the total AOD (Fig. 6) on 15 October, whereas the next day, the AOD data (only available at noon because of the obscuration by the ash layer), indicate that the coarse mode dominates in accordance with those reported by Papayanis et al. (2012) over Europe after the Eyjafjallajökull eruption.

The diurnal evolution of the CN, eBC, PPAH and the extinction coefficient on 17 October is consistent with the typical urban plume, initially in a shallow stable boundary layer followed by a well-developed convective boundary layer. This is indicative that the ash plume has either been advected out of the region or it has been mostly deposited onto the surface.

**NHESSD**

doi:10.5194/nhess-2015-311

**Aerosol properties and meteorological conditions in the city of Buenos Aires**

A. G. Ulke et al.

Discussion Paper | Discussion Paper | Discussion Paper | Discussion Paper

**NHESSD**

doi:10.5194/nhess-2015-311

**Aerosol properties and meteorological conditions in the city of Buenos Aires**

A. G. Ulke et al.

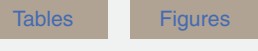

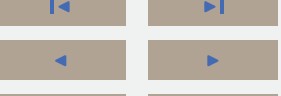



# 5  Discussion

Four months after the initial eruption of the PCCVC, the activity of the volcanic complex was weak and the ash column relatively shallow during the period studied here, so that there was little chance of fresh emissions reaching Buenos Aires in October. The observed ash plume and the ash deposited on the runways of the city airports on 16 October has to be associated with a resuspension event of previously deposited volcanic ash in northern Patagonia. The synoptic analysis and the thermodynamic soundings of the days prior to the arrival of the ash plume in Buenos Aires are all consistent with the hypothesis of advection after an event of ash resuspension. In order to test this hypothesis we use the HYSPLIT model with the "dust storm" module, as detailed in Sect. 2.3. The simulations were initialized on 13 October and performed for a total of 96 h. We present here only a few selected time periods from the simulations that coincide with the overpass of the CALIPSO satellite in order to perform a qualitative analysis of the results. In particular, we show here the results for 14 October at 18:00 UTC (Fig. 7a–c), 15 October at 06:00 UTC (Fig. 7d–f) and 16 October at 18:00 UTC (Fig. 7g–i).

Figure 8 includes for each selected period a map that indicates the trajectory, the total attenuated backscatter (TAB) at 532 nm and the derived aerosol subtype.

The TAB signal strength has been color coded such that cirrus clouds appear in gray/red/yellow colors and mid- and low altitudes clouds in white/gray/red colors. Aerosols show up as green/yellow/orange/red colored features.

Based on analyses of the CALIOP data aiming to discriminate clouds from aerosols, Liu et al. (2009) found that clouds have a bimodal distribution centered, respectively, at $\sim 0.1$ and $\sim 0.01\,\mathrm{km}^{-1}\,\mathrm{sr}^{-1}$ of attenuated backscatter and $\sim 0.95$ and $\sim 1$ of color ratio while aerosols have a single-mode distribution centered at $\sim 0.003\,\mathrm{km}^{-1}\,\mathrm{sr}^{-1}$ of attenuated backscatter and $\sim 0.45$ of color ratio. There is a small overlap region mainly seen between 0.004 and $0.01\,\mathrm{km}^{-1}\,\mathrm{sr}^{-1}$ for TAB and 0.5–0.9 for the color ratio. Dust aerosols usually have large backscatter color ratios, due to their large size. The

magnitudes of depolarization ratios for cirrus and dense water clouds are, respectively around 0.5–0.6 and 1. These values are larger than those for aerosols (from ∼ 0 to 0.1). Dust normally has large depolarization ratios due to their asphericity (∼ 0.2–0.4). In their aerosol subtyping algorithm, Omar et al. (2009) found that although the distributions of total attenuated color ratios are not identical, the mean values for the types dust, smoke and polluted dust, are centered around 0.5.

Focusing on the detection of volcanic ash, Winker et al. (2012) found a magnitude of the attenuated backscatter of the ash layer varying from $3.5 \times 10^{-3}$ to $5 \times 10^{-3}$ km$^{-1}$ sr$^{-1}$, a magnitude of the volume depolarization around 0.3–0.4, and a magnitude of the attenuated color ratio between 0.5 and 0.7.

Vernier et al. (2013) showed that the PCCVC plume was primarily made of highly depolarized ash. The volcanic ash layers exhibited color ratios near 0.5, significantly lower than unity, as is observed in ice clouds and a high volume depolarization ratio (0.3–0.4).

On 14 October, the CALIPSO revealed a zone with TAB ranging from $2.5 \times 10^{-3}$ to $4 \times 10^{-3}$ km$^{-1}$ sr$^{-1}$ located over the Patagonian steppe (∼ 39° S and 65° W, reaching 2 km height) (Fig. 8b). The vertical feature mask clearly discriminates this feature as aerosol and the aerosol subtype is dust (Fig. 8c). This zone displayed volume depolarization ratios between 0.3 and 0.4 and color ratios near 0.5 (not shown). This behavior is in agreement with the findings of Vernier et al. (2013) and Winker et al. (2012) for volcanic ash.

The HYSPLIT analysis has clearly located the emission cells at the arid zone over the Patagonian steppe. This zone was coincident with the CALIPSO overpass. The simulated horizontal plume (Fig. 7a) is consistent with the synoptic environment and the advection of the particles is carried out by strong near surface SW winds veering clockwise (cyclonic circulation for the SH) and reaching the Atlantic Ocean (not shown). The simulated particle cross sections around the time of the satellite overpass highly resemble the lidar measurements in location (∼ 35° S, 69° W) and also in the height of the plume near 3 km (compare Fig. 7b, c with Fig. 8b, c).

Discussion Paper | Discussion Paper | Discussion Paper | Discussion Paper | Discussion Paper |

**NHESSD**

doi:10.5194/nhess-2015-311

**Aerosol properties and meteorological conditions in the city of Buenos Aires**

A. G. Ulke et al.

The CALIOP lidar observations during the nighttime satellite route on 15 October depict TAB values ranging between $7 \times 10^{-4}$ and $3 \times 10^{-2}$ km$^{-1}$ sr$^{-1}$ at $\sim 40°$ S; 64.5° W to 34° S; 63° W, rising up to 3 km above ground (indicated with the white arrow in Fig. 8e).

The depolarization ratios range from 0.1 to 0.5, where the predominant values are between 0.3 and 0.4 with a heterogeneous structure. The attenuated color ratio exhibits a majority of values around 0.5–0.6, but they also show magnitudes around 1, consistent with the presence of mixed clouds (water/ice) at southern latitudes, in coincidence with the vertical feature mask (VFM) (not shown). After applying the
corresponding aerosol inversion algorithms, these features correspond to mostly dust, polluted dust and polluted continental as shown in Fig. 8f.

Although the CALIPSO ground track is tangential to the area of interest and the zone of particle emissions, a comparative analysis with the modeling results is still instructive. The horizontal plume pattern resembles the low level flow as described
in the previous section and shown in Fig. 4a. Although it is six hours later, the field configuration is quite similar. The distinguishable feature measured by CALIPSO (15 October 06:00 UTC) which contains cloud plus volcanic ash plus dust located around 38° S, 64° W previously analyzed (see paragraph above) is captured by the model although the vertical extent is underestimated.

For the daytime CALIPSO pass of 16 October, the lidar reported 532 nm total attenuated backscatter ranging from $1.5 \times 10^{-3}$ to $2 \times 10^{-2}$ km$^{-1}$ sr$^{-1}$ located at $\sim 34°$ S; 63° W to 29° S; 64.5° W, with a vertical extent reaching 2500 m (indicated with the white arrow in Fig. 8h). The gray colors are associated with clouds. The depolarization ratio is dominated by values around 0.4–0.5 (not shown) and the attenuated color ratio in
the upper part of the signature, mostly corresponds to water (1) and below this layer a mixture of values from 0.5 to 0.7 is observed in coincidence with water and unknown values (the VFM has low confidence in discriminating the features/aerosols).The aerosol layer is identified as dust and polluted dust as shown in Fig. 8i.

**NHESSD**

doi:10.5194/nhess-2015-311

**Aerosol properties and meteorological conditions in the city of Buenos Aires**

A. G. Ulke et al.

Discussion Paper | Discussion Paper | Discussion Paper | Discussion Paper | Discussion Paper

The HYSPLIT particle plume displacement follows the strong SW winds over the southern Buenos Aires province. The divergent feature of the plume (Fig. 7g) over the Atlantic Ocean is consistent with the 1000 hPa field pattern (see Fig. 4c). The model reproduces the dust plume being advected from the steppe into the Buenos Aires
Metropolitan Area and the highly populated cities of Cordoba (31°25′ S, 64°11′ W), Rosario (32°57′ S, 60°39′ W) and Santa Fe (31°38′ S, 60°41′ W). In addition to the evidence from the CALIOP measurements (not shown), the METAR information reported volcanic ash at those locations, as well.

Although the goals of our study are not to compare HYSPLIT dust module results
with other model simulations, a qualitative comparison with Folch et al. (2014) for 16 October 18:00 UTC is done. In fact, Folch et al. (2014) study was focused on the calibration of a new resuspension module.

The horizontal particle distribution depicted at Fig. 7g is quite similar to their results (see their Fig. 7d and e) obtained with the oldest resuspension schemes. Comparing
model estimates with proxy data obtained from visibility reports from the SMN, they mentioned that best results were obtained with the simplest scheme, but the horizontal fields of accumulated material were not included.

## 6  Conclusions

In June 2011, the PCCVC in southern Chile erupted explosively and released an
ash plume that caused air traffic disruption throughout the Southern Hemisphere. The eruptive activity decreased in intensity by July and low-level isolated emission events lasted until February 2012. A significant amount of PCCVC ejecta of the initial eruption was deposited on the ground in neighboring Argentina. Some of that material was later resuspended by low-level winds, similar to particles in a dust storm, and
transported far from the emission source. One such resuspension episode occurred in mid-October 2011 which impacted the city of Buenos Aires and resulted in the closure of both its airports.

**NHESSD**

doi:10.5194/nhess-2015-311

**Aerosol properties and meteorological conditions in the city of Buenos Aires**

A. G. Ulke et al.

Discussion Paper | Discussion Paper | Discussion Paper | Discussion Paper |

In this study we analyze the meteorological conditions that led to the episode of volcanic ash resuspension and its transport to the city of Buenos Aires and relate the event with the measurements of aerosol properties being carried out at Ciudad Universitaria.

⁵ The synoptic conditions supported the presence of very intense near-surface winds in northern Patagonia, where the ash had been deposited after the initial eruptions of the PCCVC. Moreover, the patterns were optimal for the transport to the region of Buenos Aires. The analysis of thermodynamic soundings indicated that the presence of the resuspended ash resulted in a drying of the lower troposphere, an effect already ¹⁰ reported in studies elsewhere as a result of the composition of the ash.

Although the instruments deployed at Ciudad Universitaria were mainly tailored for the study of urban air pollution, the analysis of the observations indicated that the parameters did not follow the typical diurnal pattern of urban pollution. The PPAH, eBC and CN showed unusual behavior and anomalous correlations not associated with ¹⁵ urban emissions. Light absorbing particles, not associated with typical urban pollution were identified from the in situ measurements on 16 October, coinciding with METAR reports of ash on the runways at both airports and very large $PM_{10}$ concentrations, exceeding the daily standard by up to 60 %, reported by the air quality network in the city. Moreover, the ceilometer measurements detected the presence of the ash ²⁰ plume, with a non-uniform vertical structure that clearly impacted the research site from midnight 15 October until the afternoon of 16 October. On the regional scale, the lidar measurements from the CALIOP satellite validated the meteorological analysis of the resuspended ash location.

We use the HYSPLIT model with the dust storm module to simulate the episode ²⁵ based on the similarity of the dynamics of ash resuspension and dust storms. The resuspension of aged volcanic ash combined with dust from the Patagonian steppe appears to be a result of a superposition of several factors:

**NHESSD**

doi:10.5194/nhess-2015-311

**Aerosol properties and meteorological conditions in the city of Buenos Aires**

A. G. Ulke et al.

- the region where ash was deposited is a recognized source of wind-blown dust due to the combination of the arid soil and quasi-permanent strong westerly winds, conditions that favor lifting mechanisms for dust.

- the climatological analysis revealed 2011 as a particularly dry year.

- the meteorological conditions on these days were characterized by intense southwesterly winds ahead of a high pressure system over the Patagonian steppe.

In general, the simulations with the HYSPLIT model captured the resuspension episode and the ash/dust plume timing and location after the surface soil type appropriate for the Patagonian steppe had been incorporated in the model. The introduction of the proper soil type in the model was crucial for producing reasonable results.

This study demonstrates the applicability of the dust storm module in HYSPLIT as an adequate tool for the simulation of ash resuspension under the conditions prevalent in Patagonia. Given the large number of volcanoes in the Patagonian Andes and the prevailing westerly winds, it is very likely that most of the ash from future eruptions will again fall over the Patagonian steppe, where conditions will be conducive to resuspension episodes that can potentially affect air traffic in the vicinity of large urban centers.

*Acknowledgements.* This research was partially funded by projects UBACyT X224, 20020100101013 and 20020130100771, ANPCyT PICT 08-1739 in Argentina and AO-LEFE-CHAT 875064 from CNRS, France. Universidad Nacional Autónoma de México (UNAM) is gratefully acknowledged for covering the costs of transportation of the equipment from Mexico City to Buenos Aires. Partial funding was also provided by grant Conacyt-Semarnat 23498 in Mexico. The National Weather Service of Argentina provided the upper air soundings and surface meteorological observations. The authors would like to thank B. Holben and the AERONET PIs for collecting the aerosol observations around the world. NCEP is acknowledged for the meteorological data. NASA is acknowledged for the CALIPSO and MODIS images and NOAA for HYSPLIT model. The Air Quality Monitoring Network of the city of Buenos

**NHESSD**

doi:10.5194/nhess-2015-311

**Aerosol properties and meteorological conditions in the city of Buenos Aires**

A. G. Ulke et al.

Discussion Paper | Discussion Paper | Discussion Paper | Discussion Paper

Aires is acknowledged for the $PM_{10}$ measurements. SERNAGEOMIN from Chile is gratefully acknowledged for providing the daily reports that describe the eruptive process. Finally, the authors are grateful to the staff of the Facultad de Ciencias Exactas y Naturales (UBA) for their invaluable help and logistic support during the campaign.

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

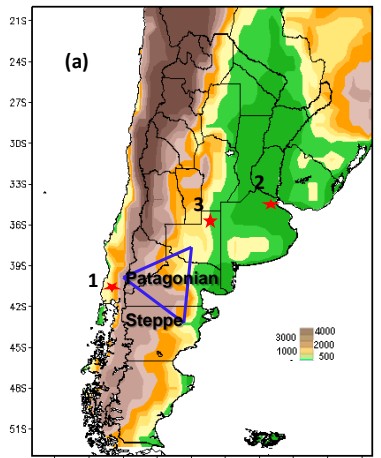

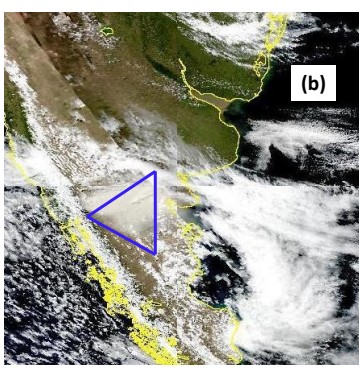

**Figure 1. (a)** Topography of Argentina and locations referred to in the text (red stars): (1) PCCVC volcano, (2) Buenos Aires City, (3) Santa Rosa City; **(b)** MODIS ESDT: MYD09, SDS name: 1 km Surface Reflectance Band 1,4,3, Sensor acquisition date: 15 October 2011 Aqua MYBGLSR, Day 2011288, Collection 005.

**NHESSD**

doi:10.5194/nhess-2015-311

**Aerosol properties and meteorological conditions in the city of Buenos Aires**

A. G. Ulke et al.

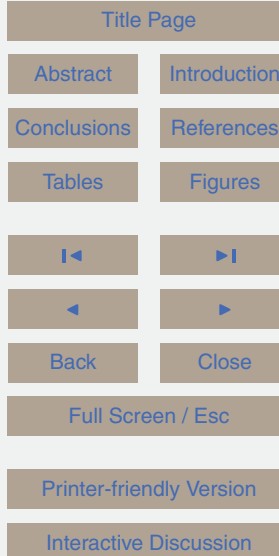

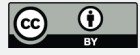

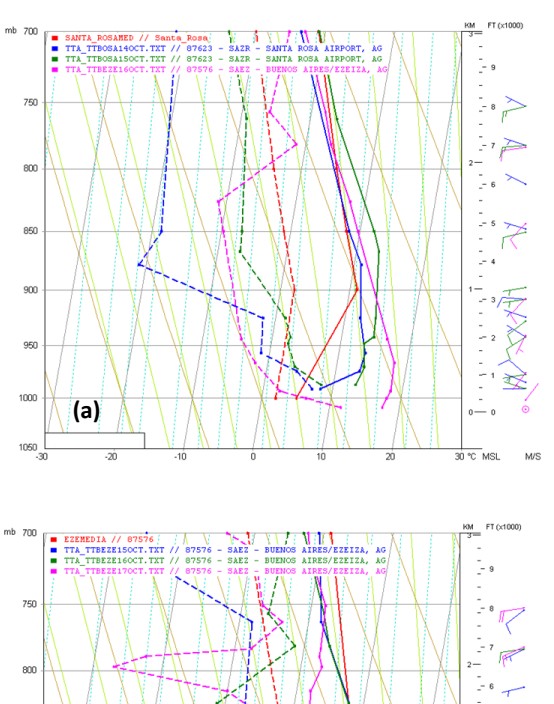

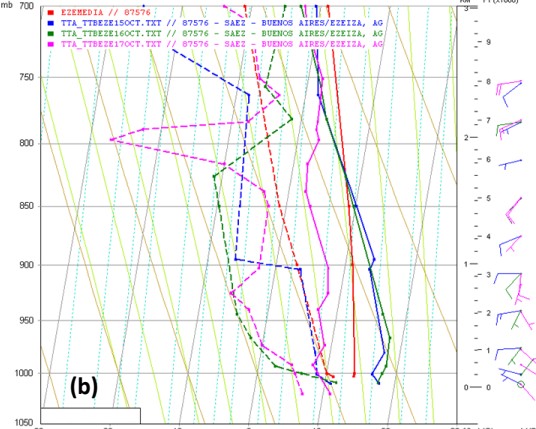

**Figure 2. (a)** Thermodynamic profiles at 12:00 UTC over Santa Rosa. Wind barbs are in $m\,s^{-1}$. **(b)** Thermodynamic profiles at 12:00 UTC over Ezeiza. Wind barbs are in $m\,s^{-1}$.

**NHESSD**

doi:10.5194/nhess-2015-311

**Aerosol properties and meteorological conditions in the city of Buenos Aires**

A. G. Ulke et al.

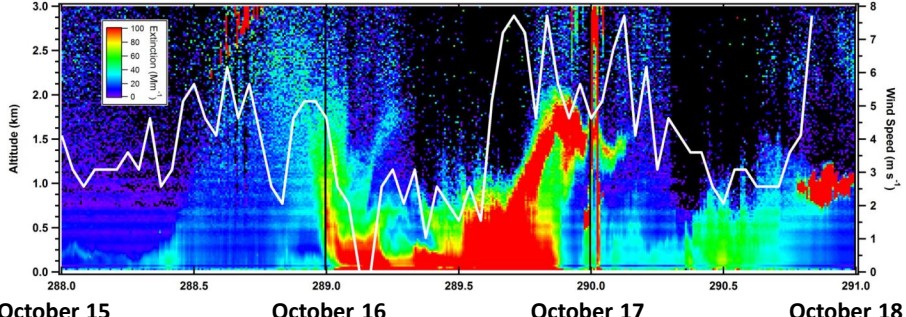

**Figure 3.** Time evolution (LST) of the extinction coefficient (Mm$^{-1}$) derived from ceilometer and surface wind speed (m s$^{-1}$) for the period from 15 October (DOY 288) to 17 October (DOY 290).



## NHESSD

doi:10.5194/nhess-2015-311

**Aerosol properties and meteorological conditions in the city of Buenos Aires**

A. G. Ulke et al.

**Figure 4.** Synoptic conditions observed at 12:00 UTC on 15 October **(a)** 1000/500 depth (shaded) and geopotential heights at 1000 hPa (contours). Arrows correspond to wind vectors at 1000 hPa. **(b)** Equivalent potential temperature at 850 hPa. Arrows correspond to wind vectors at 850 hPa. Synoptic conditions observed at 12:00 UTC on 16 October **(c)** 1000/500 depth (shaded) and geopotential heights at 1000 hPa (contours). Arrows correspond to wind vectors at 1000 hPa. **(d)** Equivalent potential temperature at 850 hPa. Arrows correspond to wind vectors at 850 hPa. Synoptic conditions observed at 12:00 UTC on 17 October **(e)** 1000/500 depth (shaded) and geopotential heights at 1000 hPa (contours). Arrows correspond to wind vectors at 1000 hPa. **(f)** Equivalent potential temperature at 850 hPa. Arrows correspond to wind vectors at 850 hPa.

# NHESSD

doi:10.5194/nhess-2015-311

**Aerosol properties and meteorological conditions in the city of Buenos Aires**

A. G. Ulke et al.

Discussion Paper | Discussion Paper | Discussion Paper | Discussion Paper | Discussion Paper

**NHESSD**

doi:10.5194/nhess-2015-311

**Aerosol properties and meteorological conditions in the city of Buenos Aires**

A. G. Ulke et al.

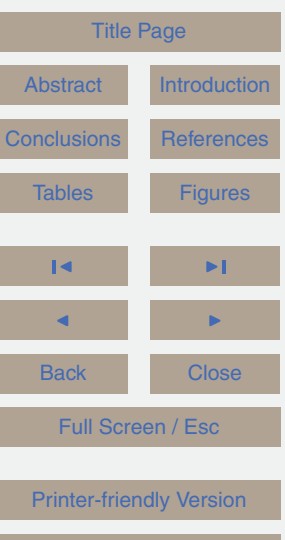



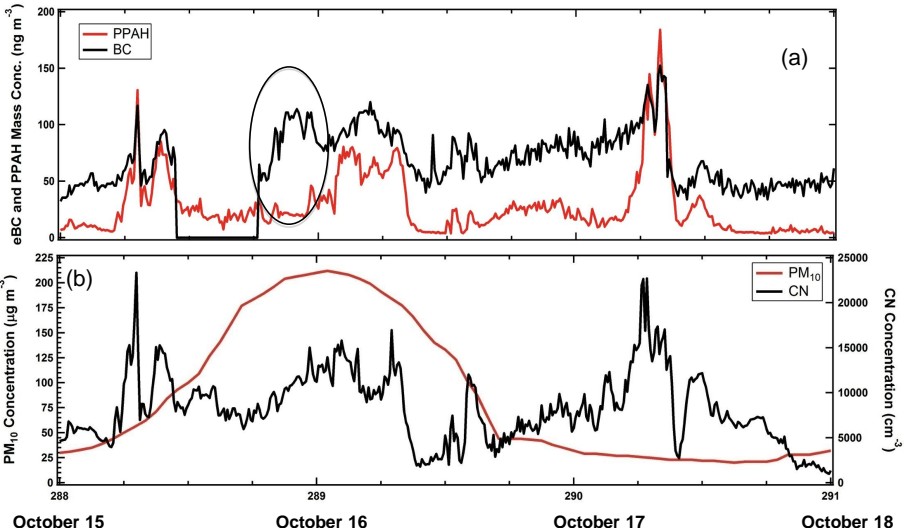

**Figure 5.** Time evolution of **(a)** PPAH (red) and equivalent Black Carbon (black). **(b)** $PM_{10}$ (red) and CN (black), for 15 October (DOY 288) to 17 October (DOY 290). Time resolution is 10 min except for $PM_{10}$ from station La Boca which is hourly. The black oval in **(a)** indicates the arrival of the resuspended ash to the research site.

Discussion Paper | Discussion Paper | Discussion Paper | Discussion Paper

**NHESSD**

doi:10.5194/nhess-2015-311

**Aerosol properties and meteorological conditions in the city of Buenos Aires**

A. G. Ulke et al.

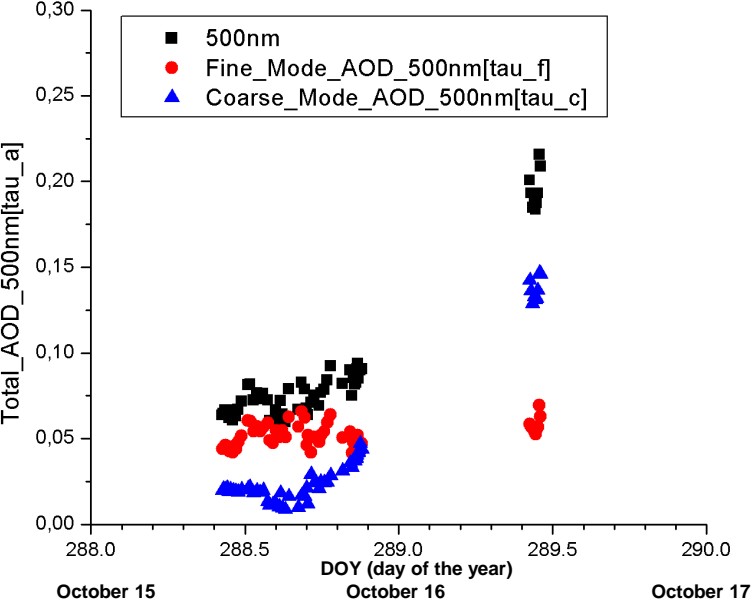

**Figure 6.** Time evolution of AOD at CEILAP Buenos Aires site for 15 October (DOY 288) to 16 October (DOY 289) in UTC time.

# NHESSD

doi:10.5194/nhess-2015-311

**Aerosol properties and meteorological conditions in the city of Buenos Aires**

A. G. Ulke et al.

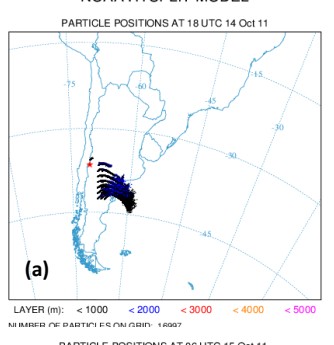

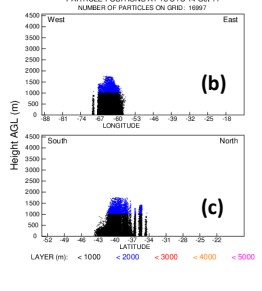

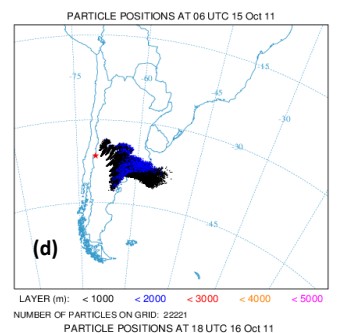

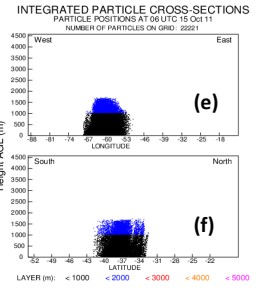

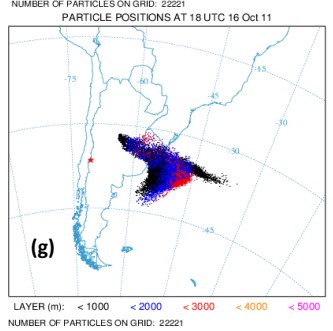

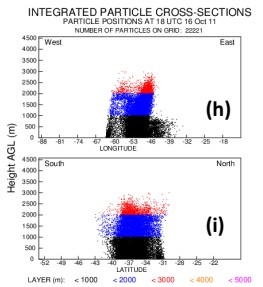

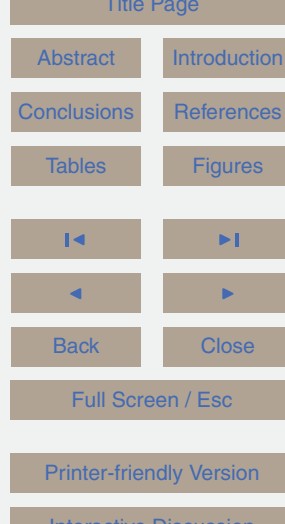



**Figure 7.** HYSPLIT simulation at 14 October 18:00 UTC. Particle positions at: **(a)** surface **(b)** W–E cross section **(c)** S–N cross section. HYSPLIT simulation at 15 October 06:00 UTC. Particle positions at: **(d)** surface **(e)** W–E cross section **(f)** S–N cross section. HYSPLIT simulation at 16 October 18:00 UTC. Particle positions at: **(g)** surface **(h)** W–E cross section **(i)** S–N cross section.

**NHESSD**

doi:10.5194/nhess-2015-311

**Aerosol properties and meteorological conditions in the city of Buenos Aires**

A. G. Ulke et al.

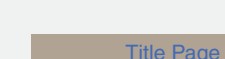

Title Page

Abstract Introduction

Conclusions References

Tables Figures

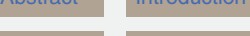

# NHESSD

doi:10.5194/nhess-2015-311

**Aerosol properties and meteorological conditions in the city of Buenos Aires**

A. G. Ulke et al.

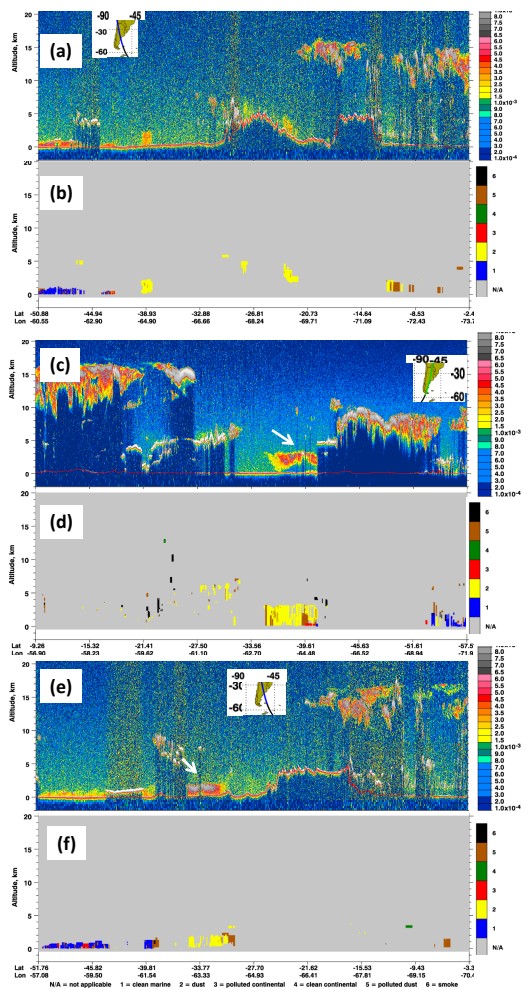

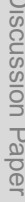

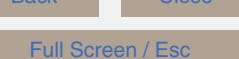

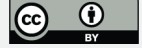

**Figure 8.** CALIPSO observations on 14 October **(a)** total attenuated backscatter at 532 nm ($km^{-1}\,sr^{-1}$). CALIPSO observations on 15 October **(c)** total attenuated backscatter at 532 nm ($km^{-1}\,sr^{-1}$). **(d)** Aerosol subtype. The ground track is included in the insert. CALIPSO observations on 16 October **(e)** total attenuated backscatter at 532 nm ($km^{-1}\,sr^{-1}$). **(f)** Aerosol subtype. The ground track is included in the insert.

Discussion Paper | Discussion Paper | Discussion Paper | Discussion Paper | Discussion Paper |

**NHESSD**

doi:10.5194/nhess-2015-311

**Aerosol properties and meteorological conditions in the city of Buenos Aires**

A. G. Ulke et al.