# Peer review of "Aerosol properties and meteorological conditions in the city of Buenos Aires, Argentina during the resuspension of volcanic ash from the Puyehue-Cordón Caulle eruption"

_Natural Hazards and Earth System Sciences, 2015_

## Referee Comment (RC1) · Anonymous Referee #1 · 17 Feb 2016

**1   General Comments**

The quantitative study of resuspended or wind blow volcanic ash is a relatively new field and due to the potential for the ash to impact both on human health and the environment it is an important area of study. In this paper the authors focus on a single resuspension event which involved the transportation of volcanic ash from the Patagonian Steppe to Buenos Aires. The authors describe the meteorological conditions leading to and resulting from the advected volcanic ash as well as looking at aerosol properties within Buenos Aires. In addition they explored whether the resuspension

event could be modelled using the HYSPLIT dispersion model with some modification to the surface properties utilised in the dispersion model.

The paper contains new observations of resuspended ash as well as discussion of the meteorological conditions leading to and resulting from the resuspension event and is, therefore, publishable. However, from a reader's perspective it feels as though the paper covers too many different observations/topics and as a result each area is only explained briefly. I think this paper could be greatly improved by some reorganisation of the text, as well as selecting topics/observations on which to focus and expanding the discussion on those topics. I've included some more detail on possible modifications below.

**2 Specific Comments**

To help the paper to flow better I would recommend that the authors move each method section next to the description of the results. For example if section 2.3 (the modelling methodology) could be moved next to section 5 (the discussion) it would be easier for readers to refer to the details of the modelling while reading the results. A similar approach could be followed with the meteorology and the different observations.

I'm not an expert on the observation techniques presented and I feel that the authors present a large number of observations but don't provide enough detail on the limitations of each observation type. I think the paper could be improved by reducing the number of observations presented but adding more detail on the detection limits, limitations, locations and interpretation assumptions for each of those observations. In addition for a number of the observation types it would be useful to see some indication of typical values for an urban area. This is particularly true in figure 5 where the peaks highlighted by the authors are modest.

Section 2.2 – 1deg by 1deg resolution seems a little coarse given the distances in-

volved. I t would be good if the authors could discuss the limitations of using such a coarse resolution meteorology. For example, the impact of a new soil scheme at higher resolution than the meteorology is going to be limited by the resolution of the meteorology particularly in the determination of surface wind speeds and stresses.

Section 2.3 – It would be helpful if the authors could provide brief details of the dust scheme in HYSPLIT as they are critical to the results presented here. Given that the authors discovered that updating the soil classification scheme was essential to the model's performance it would be useful to have more details of the reclassification. What was the resolution of the new classification? How much did it differ from the original classification? How big a difference did it make to model results presented in this paper?

**2.1 Figures**

I believe that figures should (to a certain extent) stand alone and that it shouldn't be necessary to read the detail of the text to find the figure description. Therefore I would find it extremely helpful if the authors could include descriptions of all the features in the figures in the figure captions. i.e.

Figure 1: Please could the authors include a description of the triangle in the figure caption

Figure 3: Please could the authors mention that the white line denotes wind speed in the figure caption.

Figure 6: It would be helpful to mention the cause of the gap in the data in the figure caption

Figure 7: Please could the cross-sections be marked on parts (a), (d) and (g).

In addition many of the figures contain numbers and text which are too small to read.

Please could the font size be increased so that the figures can be properly understood?

In Figure 2: the pink profile in part (a) is Ezezia not Santa Rosa and I think needs to be removed.

In Figure 5: I'm not an expert in particle observations but these peaks don't look very big. It might be helpful if longer term means could be included in the background of this figure to help demonstrate that the peaks the authors point to are indeed anomalous.

In Figure 7: The release locations in the HYSPLIT model look very gridded. In my experience of dispersion models particles are normally released evenly across any grid square where the meteorological conditions are optimum for resuspension. In this figure it looks like particles are only released from one point within each 1 degree grid cell. Although it won't change the message provided by the results, if possible the authors should re-run HYSPLIT allowing particles to be released evenly across each gridbox as this will greatly improve the look of the results. Alternatively if this isn't the case some explanation of the regular release pattern should be included in the discussion of the model results.

**3  Technical comments**

Section 3.2, Line 18: suggest replacing 500/1000 thickness with 500-1000 hPa thickness as the "/" sign is suggestive of division rather than subtraction.

---

## Referee Comment (RC2) · Anonymous Referee #2 · 18 Mar 2016

In this paper the authors analyze the resuspension episode occurred in mid-October 2011 which impacted the city of Buenos Aires and resulted in the closure of airports. Authors explore the meteorological conditions that led to the episode of volcanic ash resuspension and its transport to the city of Buenos Aires using measurements of aerosol properties carried out at Ciudad Universitaria. Moreover, they use the HYSPLIT model with the dust storm module to simulate the episode finding a good correlation. I recommend the publication after that the authors clarify some specific points:

1) Abstracts: the authors should add what are the implications of their study.

[Figure]

2) A detail description of the eruption features is lacking and should be added in the Introduction section or in a new paragraph.

3) In the Measurements and methodology, a brief description of the instrumentation and data quality should be added, even if already reported in Ulke et al. (2011) and Raga et al. (2013). Moreover, the authors should give details on the properties that were measured. Perhaps a table with the type of measurements reported in the paper and their explanation could be useful.

4) In the modelling approach, an improved description of sensitivity tests carried out in the paper will be valuable.

5) Several sentences reported in the discussion section show results plotted in figures. They should be moved to a new section and deleted from the discussion.

6) The discussion should be rewritten. Many sentences of the discussion can be moved to the Results section. A comparison with results reported in Folch et al. (2014) should be also reported.

Technical corrections

Abstract

P3L9. What airports? Two or more? Specify.

P3L9. Add the location where the thermodynamic soundings and measurements of aerosol properties were done.

P3L15. Were the reports available only for one airport?

P3L15. Add the location of the airport.

Introduction

P3L24. Add references about eruption description.

P4L12. Specify the four case studies that were analyzed.

P4L13. 'Vertical profiles of aerosol backscatter, measured with a ceilometer, clearly identified the presence of the volcanic ash'. Is it the result of this study? If yes, delete from the Introduction section.

P5L19. The moisture content is not present in the Result section. Add the graph or delete from the Introduction.

2. Measurements and methodology

P6L11. How do the authors identify volcanic ash from the dust by MODIS images? May they add some other analysis (T2-T1 difference channels? e.g. Corradini et al. (2010)) or some references from other satellite studies?

P6L13. What is it reported in this technical report? Add more data or delete the sentence.

P6L21. 'Condensation nuclei (CN) larger than approximately 50 nm'..what is the greatest size that can be detected?

P6L10. All the locations reported in the text should be added in the map (e.g. Figure 1a).

P6L15. What multiple wavelengths could the AERONET sun-photometer give?

P6L18. Already written in the previous paragraph. Again, how do the author distinguish volcanic ash from dust? Channels at 11 and 12 micron are usually used to identify volcanic ash.

P8L15. May the authors give detail on color ratio and why this type of measurement is useful?

P8L23-P9L10. Are the authors using the same method? If yes, improve the method description used in this analysis.

P9L8-9. Delete the sentence. No pertinent with the Measurements and methodology

section.

P9L20. Add the location in Figure 1a.

P9L20. Specify the meaning for METAR, SPECI, TAF, SIGMET.

P9L21.What is SMN for?

P9L22. Add Ezeiza location in Figure 1a.

P10L3. What approach are the authors using in their analysis?

P10L4-6. Why don't the authors compare data taken in their measurement station.

P10L10. What do you mean for 'the optimum setup'?

P10L10. How was the default land use file modified?

P11L4. May the authors add the main differences between the default land use file and the new one?

P12L12. How much dryer?

P13L21. What is METAR/SPECI?

4. Analysis of the measurements from the field campaign

P14L6. Add the location of the research site in Ciudad Universitaria.

P14L15. May the authors add the value of the correlation coefficient?

P15L4. The value of 240 $\mu$gm$-3$ is not reported in the La Boca station in Figure 5. Why?

P15L9. Add the hourly value maximum of PM10.

P15L26. May the authors show in the figure the "filament-like" plumes?

P16L1. Add the Aeroparque location in Figure 1a.

P16L20. What is the implication about coarse and fine mode?

5. Discussion

P17L15. Figure 7 is not well described. A description of the input run in the HYSPLIT simulation is necessary.

P17L17. Add from CALIPSO.

P17L20. May the authors highlight the area where aerosol are retrieved in Figure 8? It is not clearly visible.

P17-L28. Large size respect to what?

P17L10-L2. Those sentences should be moved from the Discussion to the Result sections.

P18L15-L19. Those sentences should be moved from the Discussion to the Result sections.

P18L22-P19L28. Those sentences should be moved from the Discussion to the Result sections.

P19L9. Explain the vertical feature mask. Add this feature in the plot.

P19L10. May the authors describe the corresponding aerosol inversion algorithms that they used?

P19L28. Figure 8e?

5. Conclusions

P20L25. How much far?

P21L10. Add references.

Figures and Tables

Figure 1a. To be improved including the bar scale. All the locations reported in the text should be added in Figure 1. May the authors identify volcanic ash with some standard techniques (e.g. Corradini et al., 2010)?

Figure 2. The size of characters are small. Explain the legend.

Figure 3. In the caption, add that the wind speed is plotted with the white line.

Figure 4. Are the Figure 4 (b), (d) and (f) necessary? I suggest to delete or improve them.

Figure 5. BC in the legend is for eBC?

Figure 6. The plot should be improved.

Figure 7. May the authors change the blue color with another colour (e.g. green?). The integrate particle cross-section could be deleted.

Figure 8. All the maps of the CALIPSO overpassing should be redone. The size of the characters in the x and y scales are small. In the caption Figure 8b is lacking.

In general, may the authors change the doy scale with UTC time?

References

Corradini, S., L. Merucci, A.J. Prata and A. Piscin, Volcanic ash and SO2 in the 2008 Kasatochi eruption: Retrievals comparison from different IR satellite sensors, J. Geophys. Res., 115, D00L21; doi:10.1029/2009JD013634, 2010.

Folch, A., Mingari, L., Osores, M. S., and Collini, E.: Modeling volcanic ash resuspension – application to the 14–18 October 2011 outbreak episode in central Patagonia, Argentina, Nat. Hazards Earth Syst. Sci., 14, 119–133, doi:10.5194/nhess-14-119-2014, 2014.

---

## Author Comment (AC1) · 1 May 2016

Dear Reviewer,

The Answer and supplementary material are 2 pdf documents and were uploaded as a Answer_R1.zip

in the option Supplement.

Best Regards,

Ana Graciela Ulke

[Figure]

Please also note the supplement to this comment:
http://www.nat-hazards-earth-syst-sci-discuss.net/nhess-2015-311/nhess-2015-311-
AC1-supplement.zip

―――――――――――――――――――

---

## Author Comment (AC2) · 1 May 2016

Dear Reviewer,

The Answer and supplementary material are 2 pdf documents and were uploaded as a Answer_R2.zip

in the option Supplement.

Best Regards,

Ana Graciela Ulke

[Figure]

Please also note the supplement to this comment:
http://www.nat-hazards-earth-syst-sci-discuss.net/nhess-2015-311/nhess-2015-311-
AC2-supplement.zip

───────────────────────────